# Non-equilibrium strategies enabling ligand specificity by signaling receptors

Andrew Goetz[1]*, Jeremy Barrios[2], Ralitsa Radostinova Madsen[3], Purushottam D Dixit[1,4]*

[1]Department of Biomedical Engineering, Yale University, New Haven, United States; [2]Department of Physics, Yale University, New Haven, United States; [3]MRC Protein Phosphorylation and Ubiquitylation Unit, University of Dundee, Dundee, United Kingdom; [4]Systems Biology Institute, Yale University, New Haven, United States

## eLife Assessment

This study presents a **valuable** finding about how receptor–ligand binding pathways with multi-site phosphorylation can show non-monotonic responses to increasing ligand affinity and to kinase activity. The authors provide **compelling** evidence through a simple ordinary differential equation model of such signaling networks with the key new ingredient of ligand-induced receptor degradation. The work will be of interest to physicists and biologists working on signal transduction and biological information processing.

**\*For correspondence:**
andrew.goetz@yale.edu (AG); purushottam.dixit@yale.edu (PDD)

**Competing interest:** The authors declare that no competing interests exist.

## Abstract
Signaling receptors often encounter multiple ligands and have been shown to respond selectively to generate appropriate, context-specific outcomes. At thermal equilibrium, ligand specificity is limited by the relative affinities of ligands for their receptors. Here, we present a non-equilibrium model in which receptors overcome thermodynamic constraints to preferentially signal from specific ligands while suppressing others. In our model, multi-site phosphorylation and active receptor degradation act in concert to regulate ligand specificity, with receptor degradation, a common motif in eukaryotes, providing a previously under-appreciated layer of control. Here, ligand-bound receptors undergo sequential phosphorylation, with progression restarted by ligand unbinding or receptor turnover. High-affinity complexes are kinetically sorted toward degradation-prone states, while low-affinity complexes are sorted toward inactivated states, both limiting signaling. As a result, network activity is maximized for ligands with intermediate affinities. This mechanism explains paradoxical experimental observations in receptor tyrosine kinase signaling, including non-monotonic dependence of signaling output on ligand affinity and kinase activity. Given the ubiquity of multi-site phosphorylation and ligand-induced degradation across signaling receptors, we propose that kinetic sorting may be a general non-equilibrium ligand-discrimination strategy used by multiple signaling receptors.

## Introduction

Signaling receptors routinely encounter a wide variety of extracellular ligands and decode their identity with remarkable precision to generate context-specific responses. This selective processing of environmental cues is essential for regulating diverse biological processes, including development, immune surveillance, and tissue homeostasis (*Cantley et al., 2014*). Failures in ligand discrimination underlie many diseases, including diabetes and cancer (*Madsen et al., 2025*; *Madsen and Vanhaese-broeck, 2020*).

A key determinant of ligand specificity in biochemical networks is the thermodynamic stability of molecular complexes, such as ligand–receptor or substrate–enzyme pairs. At thermal equilibrium, the abundance of complexes is determined by their equilibrium binding constants. This imposes a fundamental limit on specificity: high-affinity ligands are inevitably favored over lower-affinity competitors, with complex abundances scaling in proportion to their association constants.

Notably, many biochemical networks display paradoxical behaviors that cannot be explained by equilibrium affinity alone (*Clark et al., 1999*; *Coombs et al., 2002*; *Freed et al., 2017*; *Madsen et al., 2025*; *Myers et al., 2023*). For example, signaling receptors such as receptor tyrosine kinases (RTKs) and T cell receptors can produce stronger signaling outputs (phosphorylation levels) in response to intermediate-affinity ligands compared to low- and high-affinity ligands (*Coombs et al., 2002*; *Lever et al., 2014*; *Freed et al., 2017*; *Madsen et al., 2025*; *Myers et al., 2023*). Additionally, RTKs also exhibit a non-monotonic dependence between receptor activity and kinase activity (*Kiyatkin et al., 2020*; *Kleiman et al., 2011*). These observations raise a fundamental question: how do signaling receptors overcome thermodynamic constraints to achieve robust, ligand-specific responses?

A classic scheme to bypass limitations imposed by equilibrium thermodynamics is kinetic proofreading (KPR), a mechanism first proposed by *Hopfield, 1974* and *Ninio, 1975*. KPR enhances specificity of high-affinity ligands by introducing energy-consuming, irreversible steps, such as phosphorylation/dephosphorylation cycles, that amplify differences between competing ligands. KPR has been invoked in diverse systems, including DNA replication (*Hopfield, 1980*), mRNA surveillance (*Hilleren and Parker, 1999*), protein folding (*Gulukota and Wolynes, 1994*), and immune receptor signaling (*McKeithan, 1995*; *Huang et al., 2019*; *Lever et al., 2014*). Notably, while most KPR models prefer ligands with the highest affinity, it is also known that embedding KPR schemes in larger biochemical networks may allow non-monotonic dependence between ligand affinity and network activity (*Lever et al., 2014*; *Murugan et al., 2014*). However, as we will show below, these models do not capture the non-monotonic dependence between network output and kinase activity.

In this work, we present a novel non-equilibrium mechanism to achieve ligand specificity at the receptor level that relies on biologically ubiquitous signaling motifs: sequential multi-site phosphorylation and active receptor degradation. These two motifs are found in many major receptor systems, including RTKs (*Furdui et al., 2006*; *Sorkin and Goh, 2009*), G protein-coupled receptors (GPCRs) (*Koenig and Edwardson, 1997*; *Tobin, 2008*), T cell receptors (*McKeithan, 1995*; *Charpentier and King, 2021*), and interleukin receptors (*Kollewe et al., 2004*; *Cendrowski et al., 2016*). Notably, the combined role of these motifs in conferring networks with ligand and kinase specificity has not been explored.

In our model, high-affinity ligand–receptor complexes are sorted toward degradation-prone states, while low-affinity complexes repeatedly dissociate the ligand, resulting in maximal signaling output only from intermediate-affinity ligands. Notably, this ligand specificity can be tuned by varying easily controllable cellular parameters, for example, enzyme abundances. This non-equilibrium kinetic sorting mechanism explains the paradoxical non-monotonic dependence of signaling activity on ligand affinity and phosphorylation rate observed in RTKs. More broadly, given the ubiquity of the signaling motifs involved, we propose that kinetic sorting provides a general strategy for achieving ligand discrimination that is likely to be broadly used across diverse signaling networks.

## Results

### Classic KPR favors high-affinity ligands

KPR is the standard model for non-equilibrium ligand discrimination. To set the stage, we first revisited the classic KPR model originally proposed by McKeithan to explain how T cell receptors avoid activation downstream of weak ligands (*McKeithan, 1995*; *Figure 1a*; see 'Materials and methods' for equations).

In this model, ligand-bound receptors undergo a series of phosphorylation steps, with the final state $P_N$ representing the active, signaling-competent form. Importantly, ligand unbinding at any phosphorylation stage returns the receptor to the unbound state $R$. We parameterized the model using dimensionless quantities: the ligand dissociation rate $\delta = k\tau$, phosphorylation rate $\omega = k\tau$, and ligand concentration $u = L/K_D$, where $K_D = k_d/k_{on}$. Assuming saturating ligand ($u \to \infty$), the steady-state abundance of the active state is

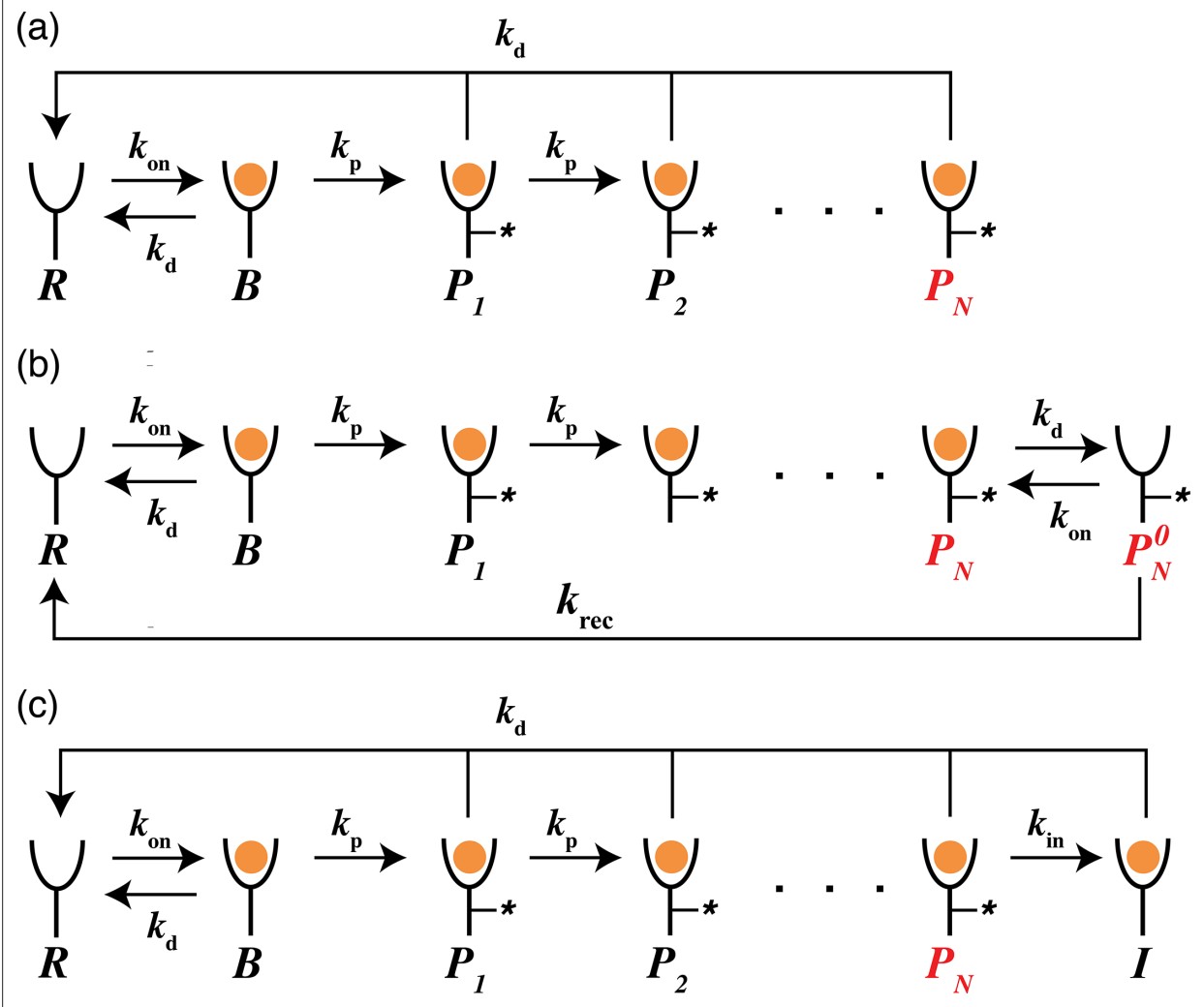

**Figure 1.** Reaction scheme of kinetic proofreading models. Chemical species and rate constants are shown in the figure. $R$ denotes ligand-free receptors, $B$ denotes ligand-bound inactive receptors, and $P_n$, $n \in [1, N]$ are phosphorylated receptors. The ultimate phosphorylated species $P_N$ (marked red) is assumed to be signaling competent. (**a**) shows the traditional model first proposed by **McKeithan, 1995**. (**b, c**) show the sustained signaling model and the limited signaling model (**Lever et al., 2014**) which introduce additional receptor states, $P_N^0$ and $I$ respectively, directly following receptor activation.

$$P_N = \frac{\omega^N}{(\omega + \delta)^N}. \tag{1}$$

As expected, increasing the phosphorylation cascade length $N$ amplifies the preference for low-dissociation (high-affinity) ligands (**Figure 2a**), reflecting the classical KPR outcome.

## Modified KPR schemes do not explain paradoxical RTK behavior

Before introducing our model, we briefly review two previously proposed extensions of receptor-level KPR that exhibit non-monotonic ligand discrimination: the sustained signaling model and the limited signaling model (**Lever et al., 2014**; **Figure 1b and c**). Both models introduce an additional state to Mckeithan's KPR scheme. The sustained signaling model adds an active but ligand-free state $P_N^0$, while the limited signaling model introduces an inactivated state $I$ downstream of $P_N$.

While both models show non-monotonic dependence of signaling activity on ligand affinity (**Lever et al., 2014**), only the limited signaling model retains this non-monotonic dependence at saturating ligand concentrations (**Lever et al., 2014**; **Figure 2b**), consistent with some paradoxical features observed in RTKs (**Freed et al., 2017**; **Madsen et al., 2025**; **Myers et al., 2023**). However, the

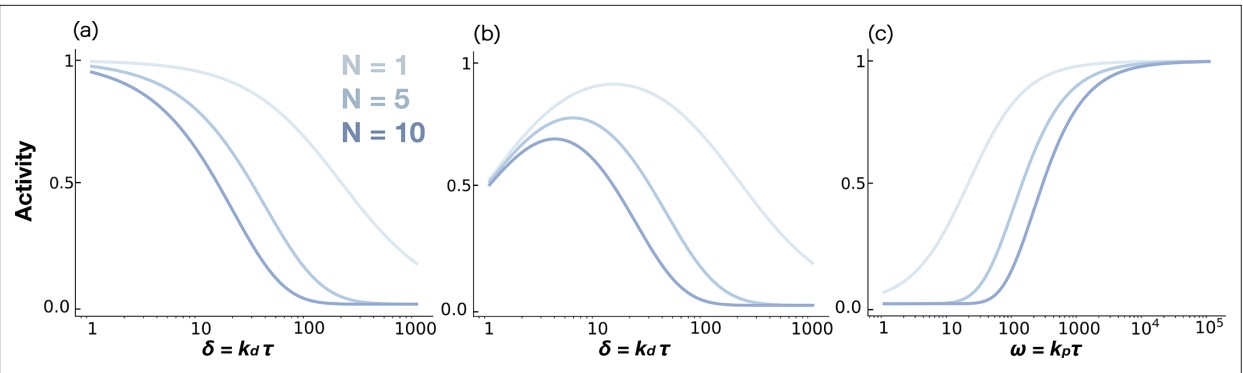

**Figure 2.** Ligand discrimination in kinetic proofreading models. (**a**) Activity $P_N$ plotted as a function of non-dimensional ligand dissociation rate $\delta$ for the traditional KPR scheme (**Figure 1a**). (**b**) Activity $P_N$ plotted as a function of non-dimensional ligand dissociation rate $\delta$ for the limited signaling model (**Figure 1b**). (**c**) The dependence of the activity on the dimensionless phosphorylation rate $\omega$ for the limited signaling model. All figures plotted for a sequence of $N = 1$, 5, and 10 phosphorylation sites.

limited signaling model fails to reproduce a second key observation in RTKs: receptor activity in this model increases monotonically with kinase activity, whereas RTK experiments show that partial kinase inhibition can paradoxically increase receptor activity (**Kiyatkin et al., 2020**; **Kleiman et al., 2011**; **Figure 2c**). Thus, these models are insufficient to explain RTK signaling dynamics.

Notably, these models neglect a key feature of many receptor signaling pathways: preferential degradation of activated receptors (**Sorkin and Goh, 2009**; **Koenig and Edwardson, 1997**; **Charpentier and King, 2021**; **Cendrowski et al., 2016**). Below, we incorporate preferential degradation in our model to investigate how it governs receptor activity.

## A kinetic sorting model integrates active receptor degradation

We build a model to study the effect of two widespread signaling motifs: sequential multi-site phosphorylation and ligand-induced receptor degradation (**Figure 3**) on ligand discrimination. In our model, receptors are delivered to the surface at a constant rate, internalized at a basal rate $k$, and degraded more rapidly when highly phosphorylated ($k_{int}^* > k_{int}$). Ligand-bound receptors undergo irreversible phosphorylation and dephosphorylation through distinct irreversible mechanisms. We note that both kinase and phosphatase are irreversible reactions carried out by separate enzymes. While their effect on the coarse-grained model of the receptor may appear reversible, it is important to note that receptor phosphorylation via ATP hydrolysis and removal of the phosphate group from the receptor corresponds to a futile cycle that does not recharge the ADP molecule to an ATP molecule.

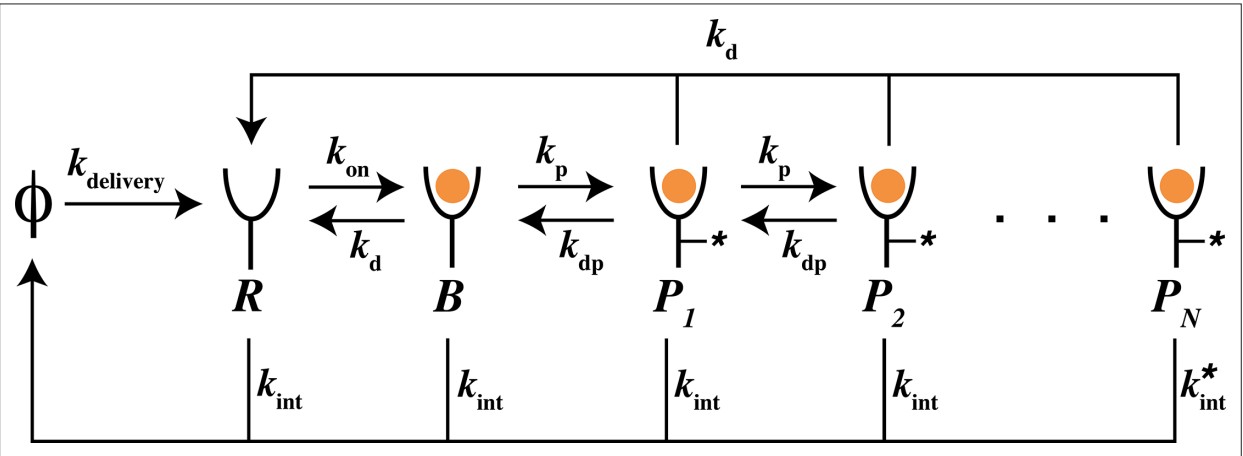

**Figure 3.** Reaction scheme of kinetic sorting model. Chemical species and rate constants are shown in the figure. $R$ denotes ligand-free receptors, $B$ denotes ligand-bound inactive receptors, and $P_n, n \in [1, N]$ are phosphorylated receptors. $\phi$ represents an implicit source and sink, corresponding to receptor delivery and internalization, respectively. It does not denote a physical chemical species.

In addition to the previously defined dimensionless parameters, we define the dimensionless active receptor degradation rate, $\beta = k^*_{\mathrm{int}}/k_{\mathrm{int}}$, and the relative rate of dephosphorylation, $\rho = k_{\mathrm{dp}}/k_{\mathrm{p}}$. A key feature of our model is that all phosphorylated species are signaling competent. Indeed, in many signaling pathways all phosphorylation sites on the receptor *Schulze et al., 2005*; *Tobin, 2008*; *Kollewe et al., 2004*; *Lemmon and Schlessinger, 2010*; *Latorraca et al., 2020* have downstream effects. Therefore, we define the net activity $A_n$ of phosphorylation site $n$ as all receptor states where the site $n$ is phosphorylated: $A_n = \sum_{m \geq n} P_m$.

## Parameter ranges

To ensure that the phenomena captured by our model are relevant to real signaling networks, we selected ranges for the dimensionless parameters based on direct experimental measurements and model fits. Importantly, many of these kinetic processes have comparable rates across diverse receptor systems (*Koenig and Edwardson, 1997*; *Subtil et al., 1994*; *Liu et al., 2000*). Specifically, basal receptor internalization occurs at rates of $k_{\mathrm{int}} \approx 10^{-4}$–$10^{-3}, \mathrm{s}^{-1}$ (*Wiley, 2003*), while active receptor internalization is typically faster, at $k^*_{\mathrm{int}} \approx 10^{-3}$–$10^{-2}, \mathrm{s}^{-1}$ (*Wiley, 2003*; *Lyashenko et al., 2020*). Ligand dissociation rates typically fall in the range $k_{\mathrm{d}} \approx 10^{-2}$–$10^{-1}, \mathrm{s}^{-1}$ (*Chen et al., 2009*; *Lyashenko et al., 2020*), and receptor phosphorylation ($k_{\mathrm{p}}$) and dephosphorylation ($k_{\mathrm{dp}}$) occur at $\sim 10^{-1}$–$10^0, \mathrm{s}^{-1}$ (*Kleiman et al., 2011*; *Chen et al., 2009*; *Lyashenko et al., 2020*). For EGFR, equilibrium dissociation constants range from $\sim 0.1, \mathrm{nM}$ for the high-affinity ligand Betacellulin to $\sim 25, \mathrm{nM}$ for the low-affinity ligand AREG (*Hu et al., 2022*; *Macdonald-Obermann and Pike, 2014*). Based on these values, we set the following ranges for dimensionless parameters: $\beta = k^*_{\mathrm{int}}/k_{\mathrm{int}} \in [1, 100]$, $\rho = k_{\mathrm{dp}}/k_{\mathrm{p}} \in [0.01, 100]$,

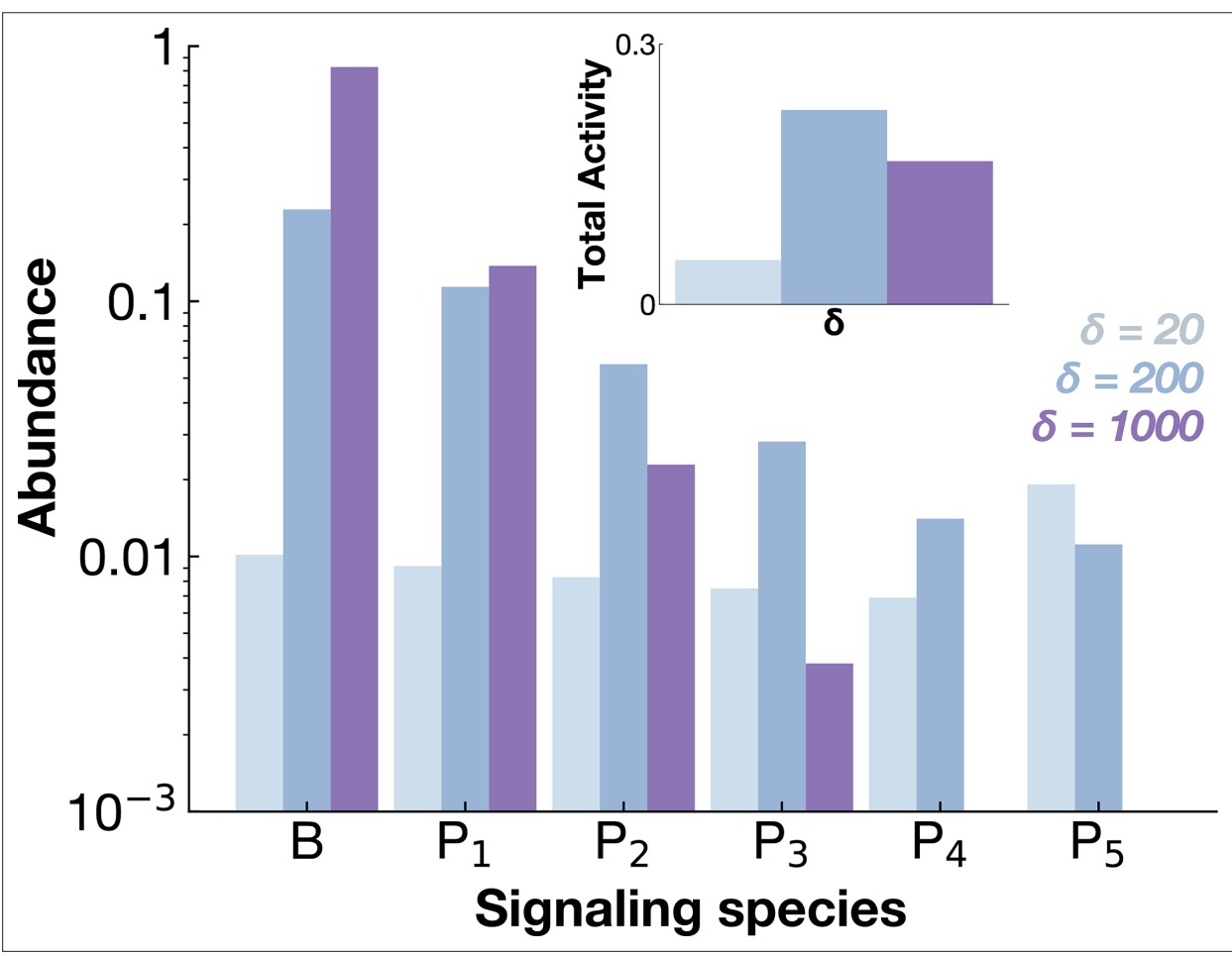

**Figure 4.** Kinetic sorting of receptor species. Abundances of network species $B$ (ligand bound inactive receptor) and $P_n, n \in [1, 5]$ for a signaling receptor with $N = 5$ phosphorylation sites. Abundances are shown for ligands of three different affinities. The inset shows the activity of the first phosphorylation site $A_1$. Species abundances below $10^{-3}$ are not shown.

$\omega = k_{dp}/k_{int} \in [1, 1000]$, and $\delta = k_d/k_{int} \in [1, 1000]$. Finally, the number of phosphorylation sites with known functional roles typically ranges from 5 to 25 (*Schulze et al., 2005*). These broad ranges comfortably encompass experimentally measured estimates. Unless otherwise specified, our default parameter values are $\delta = 20$, $\omega = 200$, $\rho = 0.01$, $\beta = 50$, and $N = 10$.

Before examining how phosphorylation levels depend on model parameters, we illustrate the mechanism of kinetic sorting of receptor states, which tunes ligand specificity beyond pure thermo-dynamic preference, using a simple example. To that end, we consider a signaling network with $N = 5$ phosphorylation sites interacting with three ligands of distinct affinities—high, medium, and low. We assume the dissociation rates for these ligands are $\delta_H = 20$, $\delta_M = 200$, and $\delta_L = 1000$, respectively. In order to compare our model with the aforementioned paradoxical experimental observations which have been performed at saturating ligand concentration, we take the limit $u \to \infty$.

*Figure 4* shows that low-affinity ligands ($\delta_L = 1000$) predominantly sort receptors toward the inactive state $B$ and early phosphorylation states $P_n, n \sim 1$ as frequent ligand unbinding prevents progression to later phosphorylation states. This behavior resembles the traditional KPR mechanism described by *McKeithan, 1995*. In contrast, receptors bound to high-affinity ligands are sorted toward later phosphorylation states, which mark them for enhanced degradation. Here, similar to traditional KPR, the fraction of receptors reaching the final phosphorylation state is highest for high-affinity ligands. Yet, the overall receptor pool is reduced due to ligand-induced degradation, lowering net phosphorylation

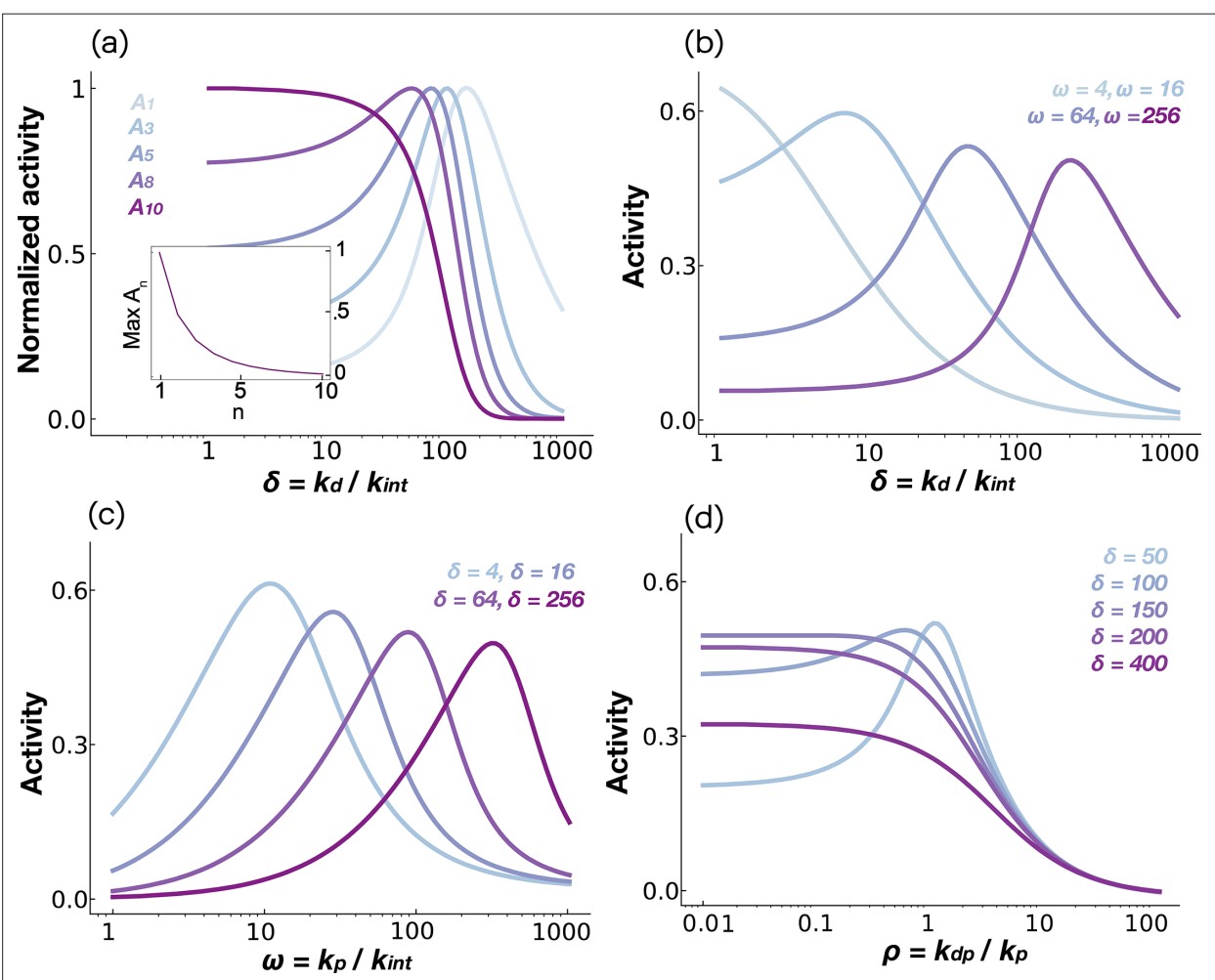

**Figure 5.** Kinetic sorting model predicts ligand specificity. (**a**) The activity $A_n$ of the $n$ phosphorylation site as a function of dimensionless dissociation rate $\delta$. The activity is normalized to the maximum activity. The maximum $A_n$ as a function of $n$ is shown in the inset. (**b**) Activity of the first phosphorylation site $A_1$ plotted as a function of the dissociation rate $\delta$ for different values of the phosphorylation rate $\omega$. (**c, d**) Activity of the first phosphorylation site $A_1$ plotted as a function of phosphorylation rate $\omega$ (dephosphorylation rate $\rho$ in panel **d**) for different values of the dissociation rate $\delta$.

activity. Strikingly, receptors bound to intermediate-affinity ligands ($\delta_M = 200$) are sorted toward intermediate phosphorylation states, resulting in maximal phosphorylation output. Below, we show how kinetic parameters govern the ability of the network to overcome thermodynamic preference and acquire ligand specificity.

## Early phosphorylation sites show ligand specificity

*Figure 5a* illustrates how total phosphorylation activity at each site, $A_n, n \in [1, N]$ varies with ligand dissociation rate $\delta$. We note that the activity of the $n$ site is given by the total concentration of all species that have the $n$ site phosphorylated; $A_n = \sum_{i=n}^{N} P_n$. We find that early phosphorylation sites ($n \sim 1$) exhibit maximal activity at intermediate values of $\delta$ while both high- and low-affinity ligands suppress net receptor phosphorylation. Our model predicts that this ligand specificity diminishes for later sites, where outputs increasingly resemble traditional KPR, which favors high-affinity ligands.

To examine how model parameters shape ligand specificity, we focused on the activity at the first phosphorylation site, $A_1$, which exhibits the strongest discriminatory behavior (*Figure 5a*). As shown in *Figure 5b*, achieving ligand specificity at high dissociation rates $\delta$ requires sufficiently high phosphorylation rates $\omega$. Notably, our model captures a puzzling observation from EGFR signaling: the high-affinity ligand EGF produces lower/comparable steady-state phosphorylation compared to lower-affinity ligands such as Epigen and Epiregulin (*Freed et al., 2017*; *Myers et al., 2023*; *Madsen et al., 2025*). Experimental estimates place the basal EGFR internalization rate at $k_{\text{int}} \approx 1.3 \times 10^{-3}, \text{s}^{-1}$ (*Chen et al., 2009*), the EGF dissociation rate at $k_d \approx 3 \times 10^{-2}, \text{s}^{-1}$ (*Chen et al., 2009*), and the phosphorylation rate at $k_p \approx 10^{-1} - 10^0, \text{s}^{-1}$, yielding $\delta_{\text{EGF}} \approx 10 - 20$ and $\omega_{\text{EGFR}} \approx 100 - 1000$. Low-affinity ligands such as Epigen (EPGN) and Epiregulin (EREG) have equilibrium dissociation constants about 10-fold higher than EGF (*Hu et al., 2022*), corresponding to $\delta_{\text{EPGN}} \approx \delta_{\text{EREG}} \approx 100 - 200$. The effective degradation rate of fully activated receptors is estimated to be 10–50 times higher than that of inactive receptors (*Lyashenko et al., 2020*), implying $\beta = 50$. Under these conditions, our model predicts a switch in phosphorylation levels: as $\delta$ increases from $\delta_{\text{EGF}}$ to $\delta_{\text{EPGN}}$, receptor phosphorylation increases—reversing the expectation based purely on thermodynamic affinity. This effect arises because EGF-bound receptors are efficiently sorted toward degradation-prone states compared to those bound to lower-affinity ligands.

Our model also explains another paradox in EGFR signaling. Experimental studies have shown that EGF-stimulated receptors exhibit higher steady-state phosphorylation when kinase activity is partially inhibited (*Kiyatkin et al., 2020*; *Kleiman et al., 2011*). As shown in *Figure 5c*, at low $\delta$ values (e.g., $\delta = 16$), decreasing the phosphorylation rate $\omega$ from levels typical of EGFR ($\omega_{\text{EGFR}} \approx 100 - 1000$) paradoxically increases overall receptor phosphorylation. A similar effect is observed when receptor dephosphorylation is enhanced (*Figure 5d*). Importantly, our model makes a testable prediction: the reversal of thermodynamic preference observed between EGF and EPGN/EREG will disappear when kinase activity is mildly suppressed (see, e.g., the curves for $\omega = 256$ and $\omega = 16$ over $\delta \in [10, 100]$), such as by treatment with low doses of the kinase inhibitor gefitinib (*Herbst et al., 2004*). This non-monotonic

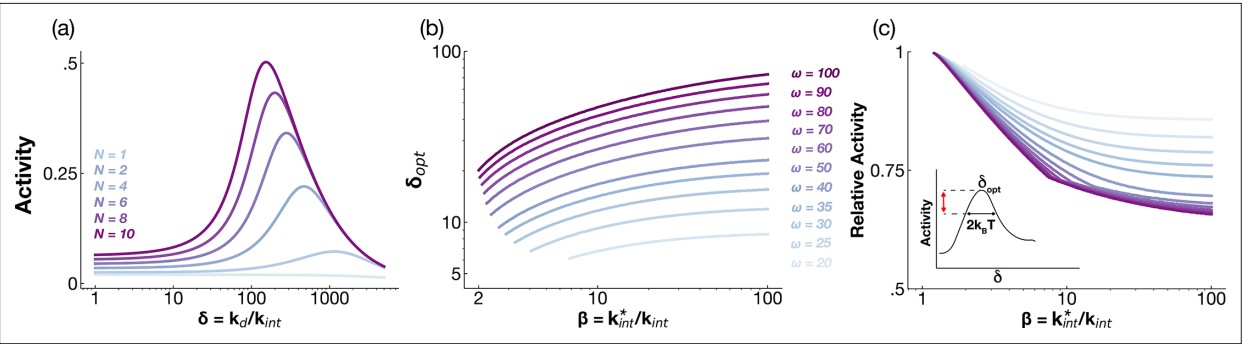

**Figure 6.** Multiple phosphorylation sites and receptor degradation dictate ligand specificity. (**a**) Activity of the first phosphorylation site, $A_1$, as a function of the dissociation rate $\delta$ for signaling networks with different number of phosphorylation sites. (**b**) The optimal dissociation rate $\delta$ that leads to maximum phosphorylation activity as a function of dimensionless degradation rate $\beta$ for different values of $\omega$. $\delta$ is shown only if $\delta_{\text{opt}} \in [1, 1000]$. (**c**) The relative activity of a ligand with dissociation rate that differs by $k_B T$ compared to $\delta$ plotted as a function of $\beta$ for different values of $\omega$ (see inset). Of the two ligands that differ in stability by $k_B T$, the ligand exhibiting maximum activity is considered.

trend may help prevent cells with abnormally high kinase activity from becoming constitutively active, thereby preserving their sensitivity to extracellular cues.

## Multi-site phosphorylation and ligand-induced degradation are both essential for ligand specificity

To assess the importance of sequential multi-site phosphorylation on ligand specificity, we analyzed $A_1^N$, the phosphorylation of the first site for signaling networks with $N$ phosphorylation sites. **Figure 6a** shows that multi-site phosphorylation is essential to endow signaling networks with ligand specificity and ligand-induced receptor degradation alone is not sufficient. This is because the non-monotonic preference for intermediate affinity ligands arises only when the receptors can be sorted among multiple phosphorylation sites: earlier ones for low-affinity ligands and later ones for high-affinity ligands.

To assess how receptor degradation shapes ligand specificity for a multi-site phosphorylation network, we examined how altering receptor turnover influences model behavior. As shown in **Figure 6b**, the optimal dissociation rate $\delta_{\mathrm{opt}}$, which maximizes receptor phosphorylation levels, increases with ligand-induced degradation rate $\beta$. Crucially, this optimal $\delta_{\mathrm{opt}}$ emerges only when receptor degradation is strong ($\beta \gg 1$). These predictions can be tested by blocking receptor degradation, for example, via mutation of ubiquitination sites (**Gerritsen et al., 2023**).

To quantify ligand specificity, we computed receptor phosphorylation in response to ligands differing by at least one $k_{\mathrm{B}}T$ in binding free energy from the optimal ligand. **Figure 6c** shows that as $\beta$ increases, phosphorylation downstream of suboptimal ligands (red line in inset) declines relative to the optimal ligand. This enhanced specificity is further amplified by increasing kinase activity $\omega$.

These results show that both multi-site phosphorylation and ligand-induced degradation are key features controlling ligand specificity in our kinetic sorting mechanism.

## Discussion

Cells face the formidable task of decoding multiple chemically distinct extracellular signals to generate appropriate, context-specific responses. This challenge is especially acute for cell surface receptors like RTKs, GPCRs, and interleukin receptors, which bind multiple cognate ligands and yet elicit distinct downstream outcomes. While equilibrium affinity provides a baseline expectation for ligand specificity, it cannot fully explain the rich and often counterintuitive behaviors observed in many signaling systems.

Here, we show that a non-equilibrium mechanism of kinetic sorting which operates through multi-site phosphorylation and active receptor degradation can explain how signaling networks achieve ligand specificity beyond equilibrium limits. In kinetic sorting, high-affinity ligand–receptor complexes are sorted toward degradation-prone states, low-affinity complexes are sorted toward inactivated states, and intermediate-affinity ligands strike the optimal balance between progression and degradation to maximize signaling. This framework explains paradoxical features observed in RTK systems, including the non-monotonic dependence of phosphorylation on ligand affinity and kinase activity. Importantly, our model predicts that early phosphorylation sites show the strongest ligand discrimination, consistent with recent experimental observations. It also makes the testable prediction that impairing receptor degradation should reduce specificity by eliminating the kinetic sorting effect. Given the ubiquity of the essential motifs of our mechanism, that is, multi-site phosphorylation and receptor degradation, we believe that kinetic sorting may be a common mechanism to modulate ligand specificity at the receptor level, potentially in addition to other mechanisms that endow signaling networks with ligand specificity, both at the receptor level (**Lever et al., 2014**) as well as in downstream signaling pathways (**Singh and Nemenman, 2017**).

In contrast to what has been shown previously for KPR models (**Coombs et al., 2002**; **Lever et al., 2014**), the kinetic sorting model also captures the non-monotonic relationship between signaling output and kinase/phosphatase activity observed in RTK systems such as EGFR (**Kleiman et al., 2011**; **Kiyatkin et al., 2020**). In these systems, partial inhibition of kinase activity paradoxically increases steady-state receptor phosphorylation, a behavior not accounted for by equilibrium models (see 'Materials and methods') or by prior non-equilibrium schemes such as the limited signaling model (**Lever et al., 2014**). This type of protective filtering can ensure that downstream signaling remains

contingent on extracellular cues and is not constitutively active, thereby preventing persistent, cue-independent activation. Such regulation could help maintain control in pathways such as those governing growth, where deregulated activity can have severe consequences. The potential benefit of this regulatory pattern suggests it could be advantageous in other signaling contexts. Consistent with this idea, non-monotonic regulation by kinase or phosphatase activity is found in other systems through distinct mechanisms (e.g., the non-monotonic effects of the phosphatase CD45 in T-cell receptor signaling, *Courtney et al., 2019*). This indicates that selective filtering based on enzymatic activity is a strategy employed in diverse biological settings. While direct evidence for the kinetic sorting mechanism remains limited to RTKs, similar filtering behavior emerges in theoretical analyses of phosphorylation–dephosphorylation cycles in more general settings (*Martins and Swain, 2013*), suggesting it may represent a broader principle of enzymatic signaling networks.

Our findings complement prior studies on mechanisms of ligand specificity that operate at thermal equilibrium, such as those described in the Bone Morphogenetic Protein (BMP) pathway (*Antebi et al., 2017*; *Su et al., 2022*; *Parres-Gold et al., 2025*). BMP signaling relies on promiscuous ligand–receptor interactions, with specificity emerging from differences in receptor abundance, binding affinity, and complex activity. In contrast, our work shows that non-equilibrium mechanisms—such as phosphorylation cycles and ligand-induced receptor degradation—can achieve ligand discrimination even for a single receptor type. Given that ligand–receptor promiscuity, multi-site phosphorylation, and receptor turnover are common features across signaling systems (e.g., in the EGFR/ErbB family; *Linggi and Carpenter, 2006*), it is likely that biological networks integrate both equilibrium and non-equilibrium strategies to achieve robust and tunable ligand specificity.

In recent years, there has been growing interest in engineering synthetic physical and chemical circuits capable of carrying out complex computational tasks, including input discrimination, classification, prediction, and the generation of multiple stable cell states (*Shakiba et al., 2021*; *Ma et al., 2022*; *Benzinger et al., 2022*; *Zhu et al., 2022*; *Floyd et al., 2024*; *Parres-Gold et al., 2025*; *Aoki et al., 2019*). Some of these synthetic strategies rely on equilibrium thermodynamics (*Parres-Gold et al., 2025*), while others exploit non-equilibrium steady states (*Floyd et al., 2024*). We propose that non-equilibrium kinetic sorting, which harnesses receptor synthesis and degradation, could provide synthetic biologists with a powerful framework for achieving precise control over molecular abundances and dynamic system behavior.

Finally, we address a major concern in non-equilibrium signaling circuits: the energetic cost of operation. Previous theoretical work has shown that free energy dissipation places fundamental constraints on the performance of signaling networks (*Bryant and Machta, 2023*; *Govern and ten Wolde, 2014*; *Lan et al., 2012*; *Mehta and Schwab, 2012*; *Qian and Reluga, 2005*; *Cao et al., 2015*; *Azeloglu and Iyengar, 2015*; *Floyd et al., 2024*; *Mahdavi et al., 2024*). These studies typically focus on futile cycles of reversible modifications such as phosphorylation or methylation. In contrast, ligand-induced receptor degradation—a central feature of many signaling networks—is a far more energy-intensive process. For example, MCF10A cells maintain approximately $10^5$ EGFR molecules on the surface (each 1,210 amino acids in length) (*Shi et al., 2016*), with a synthesis rate of about 15 receptors per second (*Lyashenko et al., 2020*), corresponding to an energetic cost of roughly $\sim 8 \times 10^4$ ATP/s (assuming 4.5 ATP per peptide bond; *Milo et al., 2010*). By comparison, EGFR dephosphorylation occurs over $\sim 15$ s (*Kleiman et al., 2011*), and only 5–10% of receptors are phosphorylated at steady state (*Shi et al., 2016*; *Feng et al., 2023*), resulting in a much lower energetic cost of $\sim 6 \times 10^2$ ATP/s for dephosphorylation. Thus, the energetic burden of receptor turnover can exceed that of reversible modification cycles by up to two orders of magnitude. These estimates suggest that, at least in eukaryotic cells where signaling proteins may turnover multiple times within cellular lifetime (*Milo et al., 2010*), non-equilibrium modification cycles are unlikely to pose a fundamental energetic limitation on the functionality of signaling networks. Here, the energetic demands of signaling networks must account for protein turnover in addition to non-equilibrium modification cycles.

## Materials and methods
### Equations for proofreading models
The equations describing species abundances in the traditional KPR model similar to that of *McKeithan, 1995* are as follows:

$$\frac{dR}{dt} = -k_{on}LR + k_d B + k_d \sum_{i=1}^{N} P_i \tag{2}$$

$$\frac{dB}{dt} = +k_{on}LR - k_d B - k_p B \tag{3}$$

$$\frac{dP_1}{dt} = k_p B - k_p P_1 - k_d P_1 \tag{4}$$

$$\frac{dP_i}{dt} = k_p P_{i-1} - k_p P_i - k_d P_i \quad \forall\, i \in [2, N-1] \tag{5}$$

$$\frac{dP_N}{dt} = k_p P_{N-1} - k_d P_N \tag{6}$$

For the limited signaling model, the dynamics of $B$, and $P_i, i \in [1, N-1]$ are identical to the traditional KPR model. The dynamics of $R$ and $P_N$ are modified as follows:

$$\frac{dR}{dt} = -k_{on}LR + k_d B + k_d \sum_{i=1}^{N} P_i + k_d I \tag{7}$$

$$\frac{dP_N}{dt} = k_p P_{N-1} - k_d P_N - k_{in} P_N \tag{8}$$

$$\frac{dI}{dt} = k_{in} P_N - k_d P_I \tag{9}$$

## Equations for the model with receptor degradation

Signaling receptors participate in a variety of complex regulatory processes, including non-linear ligand binding dynamics (*Limbird et al., 1975*; *Macdonald and Pike, 2008*), receptor oligomerization (*Mudumbi et al., 2024*; *Huang et al., 2016*), context-specific interactions with adapter proteins (*Madsen and Vanhaesebroeck, 2020*; *Feng et al., 2023*), and trafficking between cellular compartments leading to degradation (*Sorkin and Goh, 2009*; *Wiley, 2003*; *Irannejad and von Zastrow, 2014*).

While computational models that incorporate these mechanistic details are powerful tools for hypothesis generation (*Chen et al., 2010*; *Qiao et al., 2025*), they often require large-scale datasets for accurate parameterization (*Feng et al., 2023*). As an alternative, simplified models that intentionally omit certain mechanistic details can still yield deep qualitative insights, even if they cannot quantitatively reproduce experimental data.

In this study, we present such a simplified model aimed at explaining two paradoxical features of RTK signaling: (1) the non-monotonic relationship between ligand-receptor affinity and steady-state receptor phosphorylation (*Freed et al., 2017*; *Madsen et al., 2025*; *Myers et al., 2023*), and (2) the counterintuitive increase in receptor phosphorylation following mild kinase inhibition (*Kleiman et al., 2011*; *Kiyatkin et al., 2020*).

To keep the model simple and tractable, we neglect receptor recycling and oligomerization. Previously, we showed that the combined effects of endocytosis, recycling, and degradation can be captured by a single effective dimensionless parameter, $\beta$ in this study, which reflects the degradation bias of fully phosphorylated receptors compared to partially phosphorylated receptors (*Lyashenko et al., 2020*). Similarly, receptor dimerization and negative cooperativity can be abstracted into a Hill coefficient $\eta < 1$ (*Lyashenko et al., 2020*). For the phenomena explored here, including oligomerization would modify the shape of the response curves but not their qualitative behavior.

Under these assumptions, the governing equations for the model are given by

$$\frac{dR}{dt} = k_{\text{delivery}} - k_{on}LR + k_d B + k_d \sum_{i=1}^{N} P_i - k_{int} R \tag{10}$$

$$\frac{dB}{dt} = k_{on}LR - k_d B - k_p B - k_{int} B \tag{11}$$

$$\frac{dP_1}{dt} = k_p B - k_p P_1 - k_d P_1 - k_{int} P_1 \tag{12}$$

$$\frac{dP_i}{dt} = k_p P_{i-1} - k_p P_i - k_d P_i - k_{int} P_i, \quad \forall\, i \in [2, N-1] \tag{13}$$

$$\frac{dP_N}{dt} = k_{\mathrm{p}}P_{N-1} - k_{\mathrm{d}}P_N - k_{\mathrm{int}}^* P_N \tag{14}$$

All equations are solved at steady state and in the limit $u \to \infty$. All codes required to generate the figures in the manuscript can be found at https://github.com/BarriosJer0/KineticSorting (copy archived at *Barrios, 2025*).

## Equations for a model at thermal equilibrium

To confirm the role of non-equilibrium thermodynamics on ligand specificity, we consider the closest equivalent equilibrium model. The strongest requirement of an equilibrium model is that all reactions must be bidirectional. Another requirement is that microscopic reversibility or detailed balance. Specifically, ratios of rate constants around loops must equal to unity for all loops. The first requirement implies that unidirectional reactions: synthesis and degradation of receptors and the irreversible loss of activity due to ligand dissociation cannot exist in a reaction network that operates at equilibrium. The simplest equilibrium model closest to the kinetic sorting scheme is governed by the following equations:

$$\frac{dR}{dt} = -k_{\mathrm{on}}LR + k_{\mathrm{d}}B \tag{15}$$

$$\frac{dB}{dt} = k_{\mathrm{on}}LR - k_{\mathrm{d}}B - k_{\mathrm{p}}B + k_{\mathrm{dp}}P_1 \tag{16}$$

$$\frac{dP_i}{dt} = k_{\mathrm{p}}P_{i-1} - k_{\mathrm{dp}}P_i - k_{\mathrm{p}}P_i + k_{\mathrm{dp}}P_{i+1} \quad \forall\, i \in [1, N-1] \tag{17}$$

$$\frac{dP_N}{dt} = k_{\mathrm{p}}P_{N-1} - k_{\mathrm{dp}}P_N \tag{18}$$

In the above equations, we use the notation $P_0 \equiv B$.

We note that phosphorylation/dephosphorylation reactions are unidirectional non-equilibrium reactions carried out by different enzymes: phosphorylation hydrolyzes ATP to ADP and attaches a phosphate group to the receptor. In contrast, while dephosphorylation removes a phosphate group from the receptor, it does not recharge an ADP molecule back to ATP. Notably, however, this non-equilibrium nature of the phosphorylation/dephosphorylation cycle is not apparent in our coarse-grained kinetic scheme where ATP and ADP are not explicitly considered. We retain this part of the non-equilibrium model since a corresponding equilibrium model can be imagined where different sites on the receptor change conformation between an inactive and an active state and that these changes occur in a sequential manner.

Solving these equations at steady state and taking the limit $u = Lk_{\mathrm{on}}/k_{\mathrm{d}} \to \infty$, we have

$$p_i = \frac{P_i}{R_T} = \frac{\rho^{N-i}}{\sum_{i=0}^{N} \rho^i} \tag{19}$$

where $R_T$ is the total number of receptors and $\rho = k_{\mathrm{dp}}/k_{\mathrm{p}}$. Note that as expected, this equilibrium model has no dependence on ligand dissociation rate $k_{\mathrm{d}}$ at saturation, further confirming that non-equilibrium reactions are needed to endow cells with ligand specificity.

## Additional information

### Funding

| Funder | Grant reference number | Author |
| --- | --- | --- |
| National Institute of General Medical Sciences | R35GM142547 | Andrew Goetz Jeremy Barrios Purushottam D Dixit |
| Wellcome | Sir Henry Wellcome Fellowship 220464/Z/20/Z | Ralitsa Radostinova Madsen |

| Funder | Grant reference number | Author |
|---|---|---|
| UK Research and Innovation | MR/Y017439/1 | Ralitsa Radostinova Madsen |

The funders had no role in study design, data collection and interpretation, or the decision to submit the work for publication. For the purpose of Open Access, the authors have applied a CC BY public copyright license to any Author Accepted Manuscript version arising from this submission.

### Author contributions

Andrew Goetz, Conceptualization, Data curation, Investigation, Writing – original draft, Writing – review and editing; Jeremy Barrios, Software, Formal analysis, Visualization, Methodology; Ralitsa Radostinova Madsen, Conceptualization, Resources, Investigation, Methodology, Writing – original draft, Writing – review and editing; Purushottam D Dixit, Conceptualization, Resources, Data curation, Software, Formal analysis, Supervision, Funding acquisition, Validation, Investigation, Visualization, Methodology, Writing – original draft, Project administration, Writing – review and editing

### Author ORCIDs

Ralitsa Radostinova Madsen ⓘ https://orcid.org/0000-0001-8844-5167
Purushottam D Dixit ⓘ https://orcid.org/0000-0003-3282-0866

Reviewer #1 (Public review): https://doi.org/10.7554/eLife.107524.3.sa1
Reviewer #2 (Public review): https://doi.org/10.7554/eLife.107524.3.sa2
Author response https://doi.org/10.7554/eLife.107524.3.sa3

## Additional files

### Supplementary files

MDAR checklist

### Data availability

All codes are available on GitHub at https://github.com/BarriosJer0/KineticSorting (copy archived at *Barrios, 2025*).

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
