## [Editor Report · eLife Assessment]

This study presents a **valuable** finding about how receptor–ligand binding pathways with multi-site phosphorylation can show non-monotonic responses to increasing ligand affinity and to kinase activity. The authors provide **compelling** evidence through a simple ordinary differential equation model of such signaling networks with the key new ingredient of ligand-induced receptor degradation. The work will be of interest to physicists and biologists working on signal transduction and biological information processing.

---

## [Referee Report · Reviewer #1 (Public review)]

Summary:

The authors study the steady-state solutions of ODE models for molecular signaling involving ligand binding coupled to multi-site phosphorylation at saturating ligand concentrations. Although the results are in principle general, the work highlights the receptor tyrosine kinases (RTK) as model systems. After presenting previous ODE model solutions, the authors present their own "kinetic sorting" model, which is distinguished by ligand-induced phosphorylation-dependent receptor degradation and the property that every phosphorylation state is signaling competent. The authors show that this model recovers the two types of non-monotonicity experimentally reported for RTKs: maximum activity for intermediate ligand affinity and maximum activity for intermediate kinase activity.

The main contribution of the work is in demonstrating that their model can capture both types of non-monotonicity, whereas previous models could at most capture non-monotonicity of ligand binding.

Strengths:

The question of how energy dissipating, and thus non-equilibrium, molecular systems can achieve steady-state solutions not accessible to equilibrium systems is of fundamental importance in biomolecular information processing and self-organization. Although the authors do not address the energy requirements of their non-equilibrium model, their comparative analysis of different alternative non-equilibrium models provides insight into the design choices necessary to achieve non-monotonic control, a property that is inaccessible at equilibrium.

The paper is succinctly written and easy to follow, and the authors achieve their aims by providing convincing numerical solutions demonstrating non-monotonicity over the range of parameter values encompassing the biologically relevant regime.

Weaknesses:

(1) A key motivating framework for this work is the argument that the ability to tune to recognize intermediate ligand affinities provides a control knob for signal selection that is available to non-equilibrium systems. As such, this seems like a compelling type of ligand selectivity, which is a question of broad interest. However, as the authors note in the results, the previously published "limited signaling model" already achieves such non-monotonicity to ligand binding affinity. The introduction and abstract do not clearly delineate the new contributions of the model.

The novel benefit of the model introduced by the authors is that it also achieves non-monotonic response to kinase activity. Because such non-monotonicity is observed for RTK, this would make the authors' model a better fit for capturing RTK behavior. However, the broad significance of achieving non-monotonicity to kinase activity is not motivated or supported by empirical evidence in the paper. As such, the conceptual significance of the modified model presented by the authors is not clear.

UPDATE: The authors have now clarified the significance of the model in elucidating how known motifs (multisite phosphorylation and active receptor degradation) could explain the behavior, including non-monotonicity. The authors have also provided compelling arguments for the biological significance of achieving non-monotonic kinase activity response.

(2) Whereas previous models used in the literature are schematized in Figure 1, the model proposed by the author is missing (See line 97 of page 3). Without the schematic, the text description of the model is incomplete.

UPDATE: this issue has been resolved.

(3) The authors use the activity of the first phosphorylation site as the default measure of activity. This choice needs to be justified. Why not use the sum of the activities at all sites?

UPDATE: This was a non-issue. The potential misunderstanding has been mitigated by clarifications in the text.

Comments on revisions:

All issues previously identified were convincingly addressed. I have no additional suggestions.

---

## [Referee Report · Reviewer #2 (Public review)]

Summary:

In classical models of signaling network, the signaling activity increases monotonically with the ligand affinity. However, certain receptors prefer ligands of intermediate affinity. In the paper, the authors present a new minimal model to derive generic conditions for ligand specificity. In brief, this requires multi-site phosphorylation and that high-aﬃnity complexes be more prone to degrade. This particular type of kinetic discrimination allows to overcome equilibrium constraints.

Strengths:

The model is simple, and it adds only a few parameters to classical generic models. They moreover vary these additional parameters in ranges based on experimental observations. They explain how the introduction of these new parameters is essential to ligand specificity. Their model quantitatively reproduces the ligand specificity of a certain receptor. They finally provide testable prediction.

Weaknesses:

The naming of multiple variables as activity without precise definitions may be confusing to readers.

Comments on revisions:

I thank the authors for addressing my comments. One point remains regarding the naming of multiple variables as activity. Besides using other words, the authors may consider giving precise definitions of terms, e.g. by writing "We define kinase activity as the phosphorylation rate $\omega=k_p\tau$." A connection that appears only at line 204 in the present manuscript.

---

## [Author Response]

The following is the authors’ response to the original reviews.

**Reviewer #1 (Public review):**
Summary:The authors study the steady-state solutions of ODE models for molecular signaling involving ligand binding coupled to multi-site phosphorylation at saturating ligand concentrations. Although the results are in principle general, the work highlights the receptor tyrosine kinases (RTK) as model systems. After presenting previous ODE model solutions, the authors present their own "kinetic sorting" model, which is distinguished by ligand-induced phosphorylationdependent receptor degradation and the property that every phosphorylation state is signaling competent. The authors show that this model recovers the two types of non-monotonicity experimentally reported for RTKs: maximum activity for intermediate ligand affinity and maximum activity for intermediate kinase activity.The main contribution of the work is in demonstrating that their model can capture both types of non-monotonicity, whereas previous models could at most capture non-monotonicity of ligand binding.Strengths:The question of how energy-dissipating, and thus non-equilibrium, molecular systems can achieve steady-state solutions not accessible to equilibrium systems is of fundamental importance in biomolecular information processing and self-organization. Although the authors do not address the energy requirements of their non-equilibrium model, their comparative analysis of different alternative non-equilibrium models provides insight into the design choices necessary to achieve non-monotonic control, a property that is inaccessible at equilibrium.The paper is succinctly written and easy to follow, and the authors achieve their aims by providing convincing numerical solutions demonstrating non-monotonicity over the range of parameter values encompassing the biologically relevant regime.Weaknesses:(1) A key motivating framework for this work is the argument that the ability to tune to recognize intermediate ligand affinities provides a control knob for signal selection that is available to nonequilibrium systems. As such, this seems like a compelling type of ligand selectivity, which is a question of broad interest. However, as the authors note in the results, the previously published "limited signaling model" already achieves such non-monotonicity in ligand binding affinity. The introduction and abstract do not clearly delineate the new contributions of the model.

We thank the reviewer for this comment. We apologize for any unclear language on our part. The purpose of our work was not to identify the unique reaction scheme to obtain nonmonotonic dependence of network activity on ligand affinity and kinase activity. Rather, we were interested in exploring how such a dependence could arise from the interplay between two ubiquitous network motifs (multisite phosphorylation and active receptor degradation). Notably, as the reviewer later points out, previous models that incorporate only multisite phosphorylation only capture the non-monotonic dependence of network activity on ligand affinity and not kinase/phosphatase activity. We have now clarified this in the abstract (lines 14-16) and the introduction (lines 55-59).

The novel benefit of the model introduced by the authors is that it also achieves a nonmonotonic response to kinase activity. Because such non-monotonicity is observed for RTK, this would make the authors' model a better fit for capturing RTK behavior. However, the broad significance of achieving non-monotonicity to kinase activity is not motivated or supported by empirical evidence in the paper. As such, the conceptual significance of the modified model presented by the authors is not clear.

We thank the reviewer for this comment. We agree that the ability of our model to reproduce non-monotonic dependence on kinase/phosphatase activity was not sufficiently motivated in the original submission. We have now added a brief mention of the biological motivation for nonmonotonic kinase activity in the discussion (lines 229-247) to describe the potential biological significance of this behavior. In particular, non-monotonic kinase/phosphatase dependence may act as a safeguard, filtering out signaling cells with abnormally elevated kinase activity or suppressed phosphatase activity. In the presence of non-monotonic dependence on network activity, downstream signaling would remain contingent on extracellular cues, and cells with extreme kinase/phosphatase imbalances would fail to signal. This could prevent persistent, cueindependent activation, an especially important protective mechanism in pathways regulating metabolically taxing functions such as growth, proliferation, or mounting immune responses. Although direct experimental evidence for the widespread use of this mechanism is currently scarce, our motivation is supported both by the presence of similar regulatory behaviors of phosphatases which arise through distinct mechanisms (such as CD45 in T-cell receptor signaling, (Weiss, 2019)), but highlight the potential biological use of this strategy and by theoretical work on phosphorylation-dephosphorylation cycles, which demonstrates a similar effect in more general settings (Swain, 2013).

(2) Whereas previous models used in the literature are schematized in Figure 1, the model proposed by the authors is missing (see line 97 of page 3). Without the schematic, the text description of the model is incomplete.

We thank the reviewer for identifying this oversight, it has been corrected. See Figure 3 in the new text.

(3) The authors use the activity of the first phosphorylation site as the default measure of activity. This choice needs to be justified. Why not use the sum of the activities at all sites?

We thank the reviewer for this comment. We in fact study all sites (Figure 5A in the resubmitted manuscript). Notably, as suggested by the reviewer, the concentration of the first site is indeed represented by the sum of concentrations of all phosphorylated species. The concentration of the 2^nd^ site is represented by the sum of concentrations of all species except for the first one and so on (lines 153-155).

**Reviewer #2 (Public review):**
Summary:In classical models of signaling networks, the signaling activity increases monotonically with the ligand affinity. However, certain receptors prefer ligands of intermediate affinity. In the paper, the authors present a new minimal model to derive generic conditions for ligand specificity. In brief, this requires multi-site phosphorylation and that high-anity complexes be more prone to degrade. This particular type of kinetic discrimination allows for overcoming equilibrium constraints.Strengths:The model is simple, and it adds only a few parameters to classical generic models. Moreover, the authors vary these additional parameters in ranges based on experimental observations. They explain how the introduction of these new parameters is essential to ligand specificity. Their model quantitatively reproduces the ligand specificity of a certain receptor. Finally, they provide a testable prediction.Weaknesses:The naming of certain variables may be confusing to readers.

We apologize for the confusion due to unclear presentation. We have clarified our definitions throughout the manuscript.

**Reviewer #1 (Recommendations for the authors):**
(1) The abstract and introduction present the problem as if this model is solving the fundamental problem of non-monotonic dependence on ligand affinity. However, as the authors noted in their results, this problem has already been solved by a previous phosphorylation model with N-state degradation. What the authors' new model achieves is the additional experimentally observed non-monotonicity of kinase activity dependence. The abstract and introduction should be changed to reflect the actual novel contributions and also to motivate the biological significance of non-montonic kinase activity dependence.

We thank the reviewer for this comment. We apologize for any unclear language on our part. The purpose of our work was not to identify the unique reaction scheme to obtain nonmonotonic dependence of network activity on ligand affinity and kinase activity. Rather, we were interested in exploring how such a dependence could arise from two ubiquitous network motifs (multisite phosphorylation and active receptor degradation). Notably, as the reviewer later points out, previous models that incorporate only multisite phosphorylation only capture the nonmonotonic dependence of network activity on ligand affinity and not kinase/phosphatase activity. We have now clarified this in the abstract (lines 14-16) and the introduction (lines 55-59). We have also provided biological motivation behind nonmonotonic kinase activity dependance (lines 229-247).

(2) It is important to show (in the supplemental materials if needed) that the closest equilibrium analog to the model (for example, reversible rate constants from each of the activated states to an inactive state) does not achieve non-monotonicity with ligand affinity.

We have added a model in the supplementary materials that represents a detailed balance Markov chain. In the model, we imagine that ligand bound receptors undergo a series of equilibrium transitions, all characterized by the same activation and inactivation rate. We show that at saturating ligand levels, the signaling output only depends on the ratio of the activation to the inactivation rate (i.e., the thermodynamic stability of the active site) (lines 466-488).

(3) Schematics for earlier models are described in Figure 1. However, no schematic for the actual model proposed by the authors is shown. This should be added as a subpanel to Figure 1.

We thank the reviewer for identifying our omission of our model schematic. We have included our model schematic as its own figure (Figure 3).

(4) Minor: Figure 1 is referred to as Figure?? In line 97 of page 3.

We thank the reviewer for identifying this error, it has been corrected.

**Reviewer #2 (Recommendations for the authors):**
(1) There is an inconsistency between Figure 2(a) and Equation (1), it suggests that p_N is \omega^N/(\omega+\delta)^N. This makes more sense with the model defined in the supplementary material.

We thank the reviewer for identifying this error. Equation (1) has been updated to reflect the correct relationship.

(2) The figure presenting the model of the authors appears to be missing.

We thank the reviewer for identifying this error, it has been corrected (Figure 3 in the new manuscript).

(3) The authors describe phosphorylation as irreversible in the intro, but then consider reversible phosphorylation in their model, which may be confusing to readers.

We thank the reviewer for identifying this source of possible confusion. We have clarified that dephosphorylation is taken to be a distinct irreversible reaction, see lines 105 - 112.

(4) The authors reuse similar names, e.g., network activity, kinase activity, signaling activity, activity. This is confusing.

We apologize for the confusion. We note that, within the context of our model, there are important distinctions between signaling activity (the amount of signaling competent receptors) and kinase activity (value corresponding to the phosphorylation rate). We have attempted to use these different terms correctly and are happy to make clarifying corrections if there are any places where a term is misused.

(5) Several parameters are defined only in the captions of the figures, such as \beta and \rho.

We thank the reviewer for identifying this omission, we have added the definitions of beta and rho to the main text (see line 129).

(6) The sentence at line 137 lacks some words: "Below, we kinetic...".

We thank the reviewer for identifying this error, we have added the missing words (“Below, we show how kinetic…”).

(7) The sentence at line 183 lacks some words: "When kinase activity...".

We thank the reviewer for identifying this error. We have now corrected it.

(8) Figure 5 is very small.

We will work with the production team to increase the size of this figure.